# WSREB Mechanism: Web Search Results Exploration Mechanism for Blind Users

Snober Naseer [1], Umer Rashid [1], Maha Saddal [1], Abdur Rehman Khan [2], Qaisar Abbas [3,*] and Yassine Daadaa [3]

1 Department of Computer Sciences, Quaid-i-Azam University, Islamabad 45320, Pakistan; snober@cs.qau.edu.pk (S.N.); umerrashid@qau.edu.pk (U.R.); m.saddal@cs.qau.edu.pk (M.S.)
2 Department of Computer Science, National University of Modern Languages, Lahore 54000, Pakistan
3 College of Computer and Information Sciences, Imam Mohammad Ibn Saud Islamic University (IMSIU), Riyadh 11432, Saudi Arabia; ymdaadaa@imamu.edu.sa
* Correspondence: qaabbas@imamu.edu.sa

**Abstract:** In the contemporary digital landscape, web search functions as a pivotal conduit for information dissemination. Nevertheless, blind users (BUs) encounter substantial barriers in leveraging online services, attributable to intrinsic deficiencies in the information structure presented by online platforms. A critical analysis reveals that a considerable segment of BUs perceive online service access as either challenging or unfeasible, with only a fraction of search endeavors culminating successfully. This predicament stems largely from the linear nature of information interaction necessitated for BUs, a process that mandates sequential content relevancy assessment, consequently imposing cognitive strain and fostering information disorientation. Moreover, the prevailing evaluative metrics for web service efficacy—precision and recall—exhibit a glaring oversight of the nuanced behavioral and usability facets pertinent to BUs during search engine design. Addressing this, our study introduces an innovative framework to facilitate information exploration, grounded in the cognitive principles governing BUs. This framework, piloted using the Wikipedia dataset, seeks to revolutionize the search result space through categorical organization, thereby enhancing accessibility for BUs. Empirical and usability assessments, conducted on a cohort of legally blind individuals (N = 25), underscore the framework's potential, demonstrating notable improvements in web content accessibility and system usability, with categorical accuracy standing at 84% and a usability quotient of 72.5%. This research thus holds significant promise for redefining web search paradigms to foster inclusivity and optimized user experiences for BUs.

**Keywords:** blind users; web search; information exploration; usability analysis

## 1. Introduction

In the contemporary digital age, web search engines have established themselves as critical access points for online information, processing approximately 3.5 billion queries daily, a significant portion of which are centered on exploratory information seeking [1,2]. These exploratory sessions are characterized by users engaging with the search engines with complex, divergent queries aimed at broad-based learning about intricate topics [3]. This interaction typically entails users inputting keyword-based queries and consulting a series of document snippets presented in a linear list by the search engine, ranked according to their relevance to the query [4,5].

This linear interaction paradigm has not been exempted from scholarly criticism, chiefly due to its convergence tendency and a lack of alignment with the needs for diverse content exploration. The central issue lies in the fact that the search results are indexed and optimized based on offline evaluative metrics like precision and recall, which, while gauging relevance, fail to encapsulate subjective user satisfaction, particularly for blind users (BUs) [6,7]. Consequently, recent scholarly endeavors are channeling efforts towards

the development of intuitive search engines and evaluative metrics that integrate human-centric considerations [8].

Notwithstanding, the prevailing web search interfaces predominantly cater to the sighted user population, manifesting information in a linear and visually centered fashion [5]. This approach significantly hampers the approximately 75 million BUs globally from accessing and deciphering web content efficiently [7]. The linear representation necessitates BUs to engage in sequential information retrieval, prompting them to devise alternative strategies to navigate accessibility and usability barriers, including utilizing CTRL+F for manual content location, employing screen reader functionalities, or leveraging meta-information to anticipate content [9].

Therefore, it becomes imperative to acknowledge that while existing web search engines are adept at sourcing relevant web-based information, they inadvertently engender accessibility and usability impediments for BUs. The conventional display of the top-ten blue links often results in cognitive overload for BUs. In order to solve this problem, the goal of our research is to come up with a new way for BUs to look for information that takes cognition into account and allows search results to be shown in a better, more organized way. To scrutinize the efficacy of this proposed framework, we enlisted a group of legally blind individuals (N = 25) to gauge its impact on facilitating a more nuanced exploration behavior.

### 1.1. Major Contributions

The main contribution of the paper is summarized as follows.

1. Formalizing an information exploration framework considering the BU cognitive rule.
2. Categorically organizing the search results space for enhanced BU access.
3. Evaluating the proposed framework from empirical and usability perspectives on legally BUs.

Assistive technologies, such as screen readers and voice assistants, are available to aid BUs in navigating the web. However, not all websites are compatible with these tools, and their limitations may hinder a seamless experience. The cumulative cognitive load of listening to synthesized speech, processing information, and navigating can lead to mental fatigue for BUs. Hence, in this research, we formally investigate a framework that provides BUs with effective access to web content exploration.

BUs encounter a multitude of challenges when attempting to navigate the digital landscape and access online information. The predominant hurdle they face underlies the inherently linear nature of search engine results. These results, presented as ranked snippets, compel BUs to sift through information sequentially, resulting in time-consuming searches and potential disorientation. The struggle intensifies due to the deficiency of navigational aids designed with BUs in mind. Elements like buttons, menus, and images often lack proper labeling or structuring, rendering navigation a complex endeavor. Additionally, much of the online content lacks the necessary accessibility features, such as accurate heading tags, alternative text for images, and semantic markup. Consequently, screen readers—critical tools for BUs—struggle to convey content effectively. Therefore, to provide ease to the BUs, we devise a mechanism to categorically organize the search results space.

Further complicated matters pertaining to deciphering complex user interface elements using screen readers pose their own difficulties and missing labels and improper grouping can render these forms virtually unusable for BUs. Visual elements like charts and graphs—widely used to convey information—pose a significant obstacle to BUs. Without proper alternative representations, the meaning behind these visual aids is lost, hampering comprehensive understanding. Meanwhile, the overwhelming volume of web content exacerbates the challenge and these techniques must be investigated from the BU usability perspectives. Therefore, this research additionally aims to evaluate the formal BU exploration framework from the usability perspectives.

### 1.2. Paper Organization

The structure of this manuscript is delineated as follows: Section 2 encompasses a critical literature review, examining contemporary strategies prevalent in the domain. Section 3 elucidates the proposed theoretical framework in detail, forming the foundation of this study. Section 4 explains the procedural elements integral to the instantiated framework. Subsequently, Section 5 offers an analytical insight into the empirical and usability metrics applied, along with the resulting data. The penultimate Section 6 hosts an in-depth discussion, paving the way for Section 7, which culminates with concluding remarks and potential future research trajectories.

## 2. Literature Review

Blindness is a visual impairment that affects individuals' ability to perceive visual information, either partially or entirely. As a significant portion of the population, the BUs face numerous challenges in today's digital landscape. Hence, web accessibility is becoming increasingly crucial to ensure that BUs can access online information seamlessly. Understanding how BUs interact with the web is essential to fostering an online environment that adapts to their diverse needs and allows them to fully participate in the digital world. In this context, exploring the technologies that assist BUs to navigate the web effectively is becoming essential to foster an environment where BUs can participate equally. The traditional interfaces present significant challenges for the BUs when interacting with the web. The subsequent subsections briefly discuss the BU information seeking on the web and the existing accessibility technologies, along with the associated challenges.

### 2.1. BU Information Seeking

Information searching on the web is a challenging task for BUs [10]. This is due to the enormity of the information on the web and the lack of appropriate navigational support for the BUs. On the contrary, the existing web search engines, being the gateway to accessing information on the web, treat BUs similarly to sighted ones and offer no special assistance in information searching and navigation [10]. Hence, BUs are left at the discretion of navigational support from third-party assistive tools. In such a scenario, the BUs are constrained to use assistive tools [11], such as screen readers and talking software, JAWS, voice assistants, Braille, etc. The screen readers primarily convert text into synthesized speech and use an automated voice to read out the content [12]. Depending on the structure of a document, the voice may provide structural speech, including headings, links, buttons, and text, allowing BUs to navigate and interact with the information. JAWS (Job Access with Speech) is similar to screen readers with the additional functionality of Braille displays, allowing blind users to access information in Braille format [13]. The voice assistants, such as Siri, Google Assistant, Alexa, etc., can help BUs with various web-related tasks, including searching for information, setting reminders, or reading emails [14].

The BUs face a lack of assistive tools and applications. There is an immense need to develop systems that better adapt to the BU's needs and preferences [15]. While the third-party assistive tools provide an interface for accessing the information, they are incapable of effectively rendering the information best suited to the BU's cognitive capabilities. As a result, studies indicate that BUs are hesitant to use assistance due to a lack of trust in such systems.

### 2.2. BU Accessibility Standards

To overcome the structural difficulties of the content, various accessibility standards are introduced. Firstly, the Web Content Accessibility Guidelines (WCAG) and Authoring Tool Accessibility Guidelines (ATAG) introduced by the World Wide Web Consortium (W3C) provide guidelines and success criteria for making web content more accessible. This includes adding alternate text, making information navigable without a mouse, and establishing structured documents. Section 16 of the Rehabilitation Act requires [16] federal agencies to ensure that their electronic and information technology is accessible by provid-

ing appropriate captions and means to skip duplicate content. Accessible Rich Internet Applications (ARIA) ensure that the core navigational features are accessible to the BUs, such as dropdown menus and tab panels, via screen readers [17]. User Agent Accessibility Guidelines (UAAG) from W3C focus on enabling assistive technology conformance with BUs by allowing them to adjust preferences, such as speech rate and Braille display settings, to enhance their browsing experience.

However, studies report that these standards are often overlooked, and most websites do not implement these guidelines [18]. BUs express difficulty in navigating and finding the required information on the web. Moreover, a percentage of the BUs are over the age of 18 [19]. In this regard, there is an immediate need to explore a tool that can structure the content information content that adopts the BU cognitive needs and allows them to explore the information on the web effectively.

### 2.3. BU State-of-the-Art Tools

Roy et al. [20] developed a voice-activated email prototype, considering the cognitive needs of BUs. Their system operated on three basic commands: send, read, and exit to compose the email, read the inbox, and exit the program, respectively. Fayyaz et al. [21] devised an approach to reduce BU's cognitive load by presenting the summarized information in PDF tables. They used contextual information such as data types, captions, matching sentences, etc., and devised a keyboard-based navigational menu for interaction. Bukhaya et al. [22], Nair et al. [23], and Christopherson et al. [24] leveraged image processing techniques via deep learning to convert the visual information into text for subsequent processing by a text reader. Tucket et al. [25] embedded Near Field Connectivity (NFC) in academic pages preloaded with the speak command.

Zeboudj et al. [26] used the Pigeon algorithm to efficiently find relevant web pages and used resultantly retrieved web documents as pseudo-relevance feedback from the initial query. Subsequently, they extracted keywords via the Frequent Pattern Growth algorithm to determine the optimal query for reformulation. Figueroa-Gutiérrez et al. [27] proposed an architecture considering image processing techniques to automatically extract graphs under an image format, generating a description accessible to users with visual impairments. Meliones et al. [28] used the augmented voice assistance of Alexa to allow elderly BUs to generate voice commands. The system maps the request to relevant services on the web, retrieves the relevant documents, and speaks to the BUs.

However, the existing tools are concerned with enhancing the content for better accessibility by voice assistants. The summarized literature is also presented in Table 1. To the best of our knowledge, a practical investigation to restructure the information presentation mechanism for BUs considering their cognitive capabilities is yet to be undertaken. In the subsequent subsection, we briefly examine the issues and motivation for this research.

**Table 1.** Summarizing the references, approaches, and limitations of the studies mentioned in the literature review.

| Cited | Approach | Limitations |
|---|---|---|
| [10] | BUs face challenges due to lack of navigational support | Reliance on third-party assistive tools |
| [11] | Utilization of screen readers, JAWS, voice assistants | Limited effectiveness of assistive tools |
| [20] | Voice-activated email prototype | Focused on a specific application (email) |
| [21] | Reduce cognitive load using summarized information | Limited to information presented in tabular format |
| [22–24] | Leverage image processing to convert visual information | Relies on image recognition; may not cover all content |
| [25] | Embed Near Field Connectivity (NFC) for interaction | Limited to specific contexts (academic pages) |
| [26] | Pigeon algorithm for efficient web page retrieval | Focused on improving search result relevancy |
| [27] | Image processing for automatic graph description | Limited to content with graphical elements |
| [28] | Voice assistance augmentation for BUs | Primarily extends voice assistant functionality |

*2.4. Issues and Motivations*

The Internet has become the most ubiquitous technology for seeking information online. Combined with easy access to handheld devices and the ability of the web to interconnect immense amounts of information, the users' information-seeking paradigm is now relying on online information-provider services such as search engines. However, the literature has shown that 81% of Internet BU users still consider accessing online services difficult or impossible [29]. Among them, only 53% of BUs are reported to succeed in their navigation tasks on the web [30]. Hence, the web can then be a cause of exclusion for BUs. These difficulties may be explained by the inherent shortcomings of online information providers. Notably, the information interaction of BU users with online information providers is linear. This presents numerous shortcomings. The BUs has to determine the relevancy of the information content sequentially, which is time-consuming. Subsequently, information seeking in a linear search paradigm is cognitively challenging for BUs, which often results in information disorientation for BUs. Furthermore, the effectiveness of a web service is determined solely by the offline empirical evaluation measures of precision and recall, ignoring the behavioral and usability aspects of designing a search engine.

To overcome the challenges in this research, we are interested in investigating an online web search framework considering the needs of BUs. Mainly, we transformed the online information-seeking paradigm of BUs from linear to hierarchical, considering the cognitive processing capabilities of the BUs. Furthermore, we provided multimodal interaction for enhanced accessibility. Finally, we evaluated the proposed framework from the empirical and usability perspectives to determine its effectiveness by recruiting legally qualified BUs.

**3. WSREB Mechanism**

We created a WSREB mechanism to address core issues in web document exploration for BUs, i.e., non-linear navigation, accessibility, and cognition load. We instantiated the WSREB mechanism by implementing a tool that facilitates BUs to explore web search documents. Primarily, the WSREB mechanism elaborates formal representations, commencing with tree-based data models and categorical data models, component-based architecture, and SUI design. Moreover, it emphasizes accessibility and navigation of web documents while reducing the cognitive load in a non-linear and integrated way. The WSREB mechanism provides an accessible solution for BUs to explore web search results documents.

*3.1. WSREB Approach*

The WSREB approach enables the exploration of web search documents utilizing exploratory search principles. The primary goals of the approach are twofold: (i) enable BUs to convey their exploratory needs via multimodal query terms; and (ii) provide web document exploration to enhance accessibility in a non-linear way. Generally, BUs with clear search goals perform lookup activities to reach their required information. However, for exploratory needs, BUs often have ambiguous search goals.

To overcome this challenge, the proposed approach allows BUs to explore and navigate through the retrieved results, as well as refine their query as needed. Figure 1 illustrates the WSREB approach.

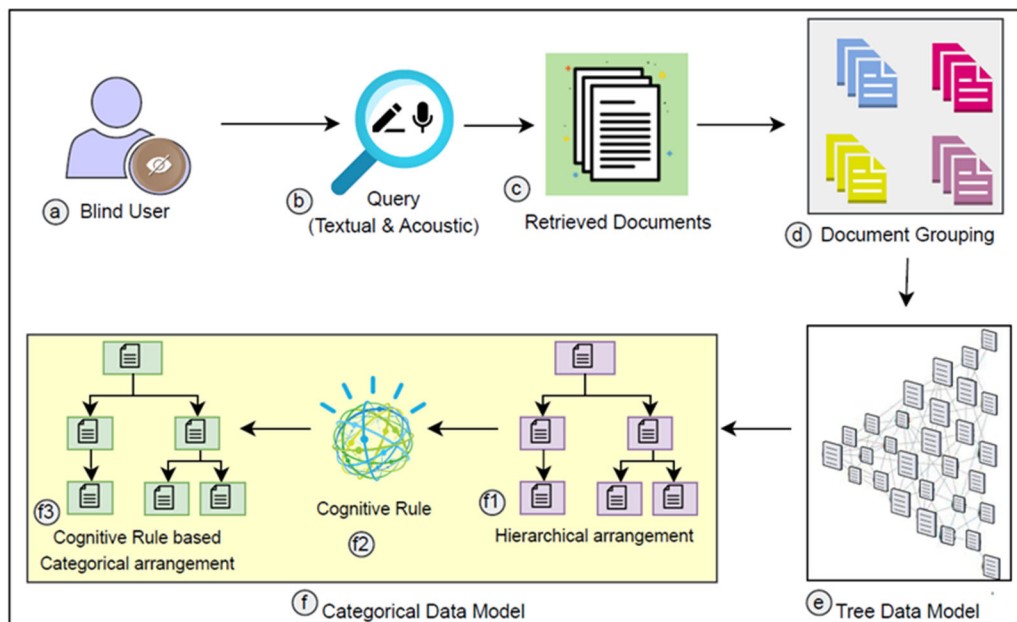

**Figure 1.** A systematic flow diagram of the WSREB approach.

Our approach allows BUs to express their information needs via multimodal query formulation. The multimodal queries comprise textual and acoustic modalities (Figure 1a,b). The capability to create multimodal queries is critical to fulfilling the exploratory needs of BUs. Textual queries aid in constructing keyword-based queries, while acoustic queries allow users to freely express complete natural language sentences via voice. The SERB system combines complex textual and auditory queries, utilizing Boolean operators (AND, OR, NOT) between the query keywords. Acoustic queries are saved and processed to assist users in the future (as depicted in Figure 1b). Typically, web search engines designed for BUs present query search results linearly, which enhances the search engines' ability to target lookup searches. A set of query-based documents is retrieved depending on the web search engines (Figure 1c). Our approach forms document groups based on the similarities (Figure 1d). These groups are unordered similarity pairs computed via statistical methods.

The mechanism of SERB provides the results in a non-linear form. For this purpose, the tree structure is introduced, which holds the results, and the results are depicted in a hierarchical form so that the exploration activity of the blind user is enhanced. The representation of results in a non-linear form helps the blind user reduce the time spent searching for the required information. The hierarchical structure converts into a categorical form. The interactive categories play a crucial role in exploring the results. Previously, interactive categories were not introduced so that the blind user could interact and search more interactively. In the representation of the categories, the Miller rule is applied, which helps the blind user. In the representation of the categories, cognitive ability is enhanced by using the rule. The blind user can easily search for information within the categories. The blind user, with or without expertise in the domain, can fulfill their information needs. The goal of the approach is to facilitate the blind user by providing a simple, clear structure for the information and a blind-friendly interface, along with interactive categories in the non-linear presentation of the search results. The shortcut keys play a major role in accessibility and interaction with the system. The shortcut keys allow the blind user to access the information; therefore, the shortcut keys are introduced to access the information more conveniently. In this way, the interaction with the blind user is more efficient and effective. The voice-over introduces a guide that guides the blind user to perform the steps and reach the destination. The structure also allows screen readers to serialize the content and read the structure in a minimum amount of time.

### 3.2. Approach Formal Algorithm Definitions

**Definition 1.** *Query:*

The BU-based query $q$ retrieves a set of documents $D = \{d_1, d_2 \ldots, d_n\}$, which contains relevant textual information. During the information-seeking journey, the user may issue multiple queries and may refer to the previously issued query. Hence, multiple issued queries can be encapsulated in a set $Q = \{q_1, q_2 \ldots, q_n\}$.

**Definition 2.** *Multimodal Queries:*

The $Q$ may comprise Textual query $Q_t$ and Acoustic Query $Q_a$. Both $Q_t$ and $Q_a$ accept multiple keywords as a set $K = k_1, k_2 \ldots, k_n$. Moreover, the query may also incorporate Boolean operators (AND, OR, NOT). Hence, the formation of a query may take the form of $k = k_1(AND \parallel OR \parallel NOT)k_n$.

**Definition 3.** *Retrieved Documents:*

The $d$ is considered as a tuple containing associated information $d = t_d, u_d, d_i$, where $t_d$ is the title of a document, $u_d$ is the URL of the document and $d_i$ is the document description. This can be formalized as $\{t_d, u_d, d_i\} \in d \in D$.

**Definition 4.** *Document Groupings:*

In $D$, each unordered pair i.e., $\{d_{n-1}, d_n\}$ underpasses through statistical methods to form multiple groups $g$ considering a threshold $\gamma$. Let a set of groups $G = \{g_1, g_2, \ldots, g_n\}$ and each $g_n$ is a tuple-containing document $g_n = \{d_1, d_2, \ldots, d_n\}$. However, all $g_n$ documents contains unique $d_n$; therefore, $g_{n-1} \neq g_n$.

**Definition 5.** *Tree Data Model:*

Formally, a document tree DT is a pair containing nodes $N$ and edges $E$ given as $T = (N, E)$. In the set of nodes $N = \{n_1, n_2 \ldots, n_n\}$, each $n$ represents the title of documents. For the set of edges $E = \{e_1, e_2 \ldots, e_n\}$, each $e$ represents weighted edges between nodes.

**Definition 6.** *Categorical Data Model:*

The categorical data model interacts with the k-array tree data model to generate hierarchies $H = \{T_1, T_2 \ldots, T_n\}$. A similarity measure $SM^y$ is applied on each $T_n$ to arrange the hierarchies in descending order of similarity to Q. This can be formalized as $H' = \{\forall (Q, T_n) || SM^y(Q, T_n) > SM^y(Q, T_{n-1})\}$, where y is a similarity threshold value.

The notable distinction in the proposed categorical model is the prevention of overlapping hierarchies since the existing literature criticizes overlapping as being difficult to interpret, especially when documents belong to multiple branches or categories within the hierarchy [31]. This can make it challenging for users to understand the structure and locate relevant documents [32]. Furthermore, as the hierarchy grows, navigation and management become more challenging [33]. Hence, the decisive hierarchical structure was chosen to ensure that the form hierarchies are easier to interpret, able to handle high-dimensional data, demonstrate simplicity and high speed, good accuracy, and the capability to produce rules for clear and understandable human classification. To further enhance navigation within the proposed model, the cognitive rule is applied to display several categories where $\{\forall T \in H' \rightarrow |T| = 7 \mp 2\}$.

---

**Algorithm 1** Algorithm for querying and retrieving documents.

---

**Input:** User Query *Q*
**Output:** Retrieved Documents *D*
**Function** Query *(Q)***:**
    **if** *modality(Q) != String* **then**
        Q = MultimodalQueries(Q);
    **end**
    query = extract_keywords(Q);
    documents = fetch_results(query)
    **return** documents
**Function** Multimodal Queries *(Qₐ)***:**
    $Q_t$ = extract_keywords($Q_a$);
    **return** $Q_t$
**Function** Retrieved  Documents *(documents)***:**
    D = [];
    **for** *document in documents* **do**
        D.append((title, URL, description = extract_metadata (*document*); D.append(tuple(title, URL, description));
    **end**
**return** D

---

### 3.3. Component-Based Architecture

Architecture is commonly defined as the "fundamental organization of a system embodied in its components, their relationships to each other and to the environment, and the principles guiding its design and evolution" [34]. The eminent feature of a component-based architectural design is the separation of concerns [35]. Therefore, the proposed mechanism employed a component-based architecture, as depicted in Figure 2. The architecture consists of five components: a web component, a query component, an information retrieval component, a tree-ranked component, a categorical component, and a blind exploration component. Each component is restricted to the assigned logic.

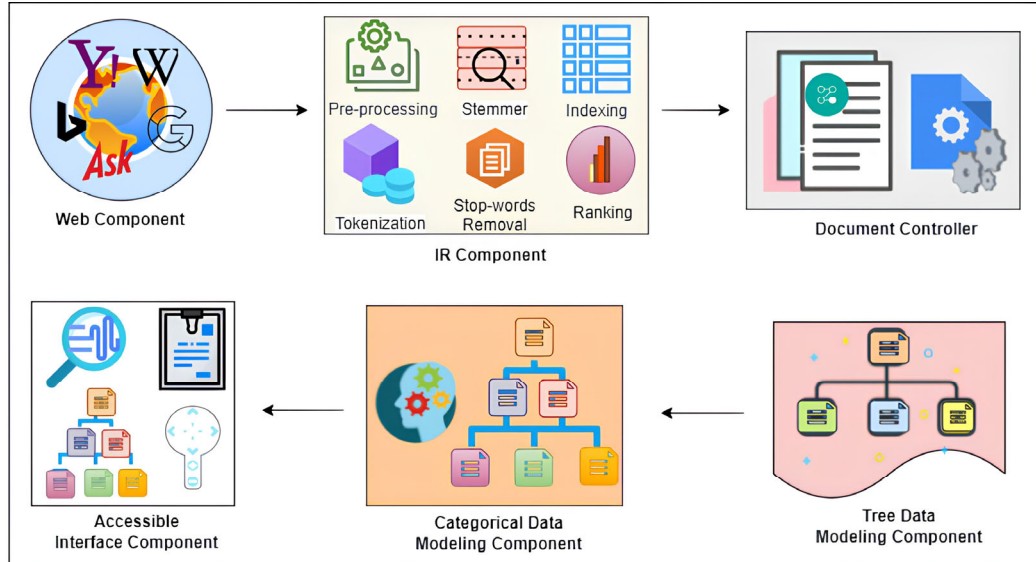

**Figure 2.** Component-based architectural representation of the proposed mechanism.

The web component accesses the textual objects by exploiting the textual modality. Considering the web documents as textual objects, this component searches the web and

archives all the documents. The IR component handles all the information retrieval tasks. The query-based retrieved documents are processed considering their title, content, and URI. The pre-processor performs parsing and term-processing while generating a logical view of the query for searching. The tokenizer creates tokens of words from the documents, which the indexer then uses to create a list by joining the tokens with the keywords. The indexer creates an inverted index while maintaining and mapping the pointers of search keywords to the documents. The indexer and term-processing correlate with the stemmer and stop-word remover. Afterward, the stemmer reduces the inflected words to their base root, while the stop-word remover eliminates the common words, computing their high frequency. Finally, the ranker is triggered, which ranks the retrieved and processed documents based on query relevancy.

---

**Algorithm 2** Algorithm for formation of tree data model.

---

**Input:** User Query *Q* and Documents *D*
**Output:** Document Tree *TD* data model
**Function** Document Grouping *(documents, query)*:

    grouping_threshold = JS(query, documents) + SD(JS);

    document_group = []; **for** *document in documents* **do**

        **if** *(JS(document) > threshold)* **then**

            document_group.append(document);

        **end**

    **end**

    **return** document_group

**Function** Tree Data Model *(documents_group, query)*:

    tree = k_array();

    _similarity_documents = JS(*query, documents_group*);

    parent_node_document = sorted_similarity_documents.pop();

    tree.add_parent(parent$_{node}$$_{d}$ocument);

    **for** *document in* sorted_similarity_documents **do**

        threshold = Mean(sorted_similarity_documents) + SD(JS);

        **if** *document == parent_node_document* **then**

            tree.append_extra_child(sorted_similarity_documents.pop());

        **end**

        **else if** *document > parent_node_document* **then**

            tree.append_left(sorted_similarity_documents.pop());

            **end**

    **Else**

    tree.append_right(sorted_similarity_documents.pop());

    **end**, **end**

    **return** tree

**Function** Categorical Data Model *(tree)*:

    min_threshold = 5;

    max_threshold = 9;

    **for** *branch in tree* **do**

        **if** *branch.levels() < min_threshold OR branch.levels() > max_threshold* **then**

            tree.remove(branch);

        **end**

    **end**

    **return** tree

---

---

**Algorithm 3** Algorithm for overall flow of data model.

---

    model. **Input:** User Query $Q$ and Documents $D$
    **Output:** Document Tree $TD$ data model
    **Function** Main *(Q)***:**
       $D$ = Retrieved Documents(Query($Q$));
       $G$ = Document Grouping($D$, $Q$);
      $DT$ = Tree Data Model($G$, $Q$);
       $C$ = Categorial Data Model($TD$);
  **return** $C$

---

The document controller applies multiple statistical computations and thresholding techniques to form groups on the received ranked documents. Each document group is a non-hierarchical group containing distinctive and unique documents. In each document group, a linear list is maintained where the document and group rankings are based on query relevancy and the IR component. The tree data modeling component converts the disjoint document groups into document hierarchies. Each disjoint document group forms a hierarchical structure, developing a parent–child relationship among the documents. This component utilizes several algorithms to form hierarchies and sustain levels of document hierarchies within a group.

The categorical data modeling component is responsible for processing the document hierarchies into categories via cognition. Each group's document hierarchy generates disjoint categories based on the hierarchy's top node and query relevancy. Furthermore, several computations re-arrange the categories in descending order and organize them on a similarity basis. Moreover, a cognitive rule is applied to categories and reorganizes them into more relevant and desired web search documents. Furthermore, each category is divided into sub-categories, and each sub-category contains multiple documents. The accessible interface component directly links with web and categorical data modeling components. The document categories are displayed along with navigational information.

The interface component has direct links with web and categorical data modeling components. The interface visualizes the web search results, highlighting the fact that this interface design is suitable for all kinds of visually impaired users. The visualization aids the other visually impaired users to interact with the web, whereas the blind users can vocally interact with it via assistive technologies. The interface component allows visually impaired or blind users to search the web via multi-modality queries, i.e., textual and acoustic queries. It visualizes the interactive categories received from a categorical component. These categories allow exploration and lookup of web search results, i.e., documents. The navigation panel enhances accessibility, allowing users to navigate and reach the documents based on their information needs.

## 4. Mechanism Instantiation

A scenario for a blind user to explore web search results instantiates the exploration mechanism. The instantiation process activates the functionality of the proposed approach using a pre-defined structure [36]. The instantiation attains the applicative scenario of the proposed mechanism, eliminating web exploration issues. The following section elaborates on the dataset, instantiation preliminaries, and implementation of the proposed exploration tool.

### 4.1. Dataset

There are various benchmark datasets available for web search documents, including MSRA, WOS, the Braille dataset (as described in Table 2), and Wikipedia, which have been extensively adopted by multiple researchers [37]. However, Wikipedia is widely used among all the datasets due to its large, consumed source of capturing knowledge. The highly dependable sources during a web search are real-time data. The significance

of Wikipedia as real-time data involves progress, data updating, analysis, and dynamic behavior. Therefore, the exploration mechanism is instantiated on a real-time Wikipedia dataset to attain all possibilities of applicative scenarios. The Wikipedia real-time dataset contains a bulk of web documents covering a diverse range of domains and topics.

**Table 2.** Archive various datasets and Tools utilized for experiments.

| No. | URL | Access Date |
| --- | --- | --- |
| 1. MSRA | https://paperswithcode.com/dataset/msra-td500 | 12 January 2023 |
| 2. WOS | https://paperswithcode.com/dataset/web-of-science-dataset | 12 January 2023 |
| 3. braille | https://www.kaggle.com/datasets/shanks0465/braille-character-dataset | 12 January 2023 |
| 4. Wikipedia | https://huggingface.co/datasets/wikipedia | 12 January 2023 |
| 5. Django | https://www.djangoproject.com/ | 13 January 2023 |
| 6. PyCharm | https://www.jetbrains.com/pycharm/ | 14 January 2023 |
| 7. VoiceAPI | https://www.twilio.com/docs/voice | 15 January 2023 |
| 8. PyLucene | https://lucene.apache.org/pylucene/ | 16 January 2023 |

*4.2. Instantiation Preliminaries*

Initially, the user issues a query. The keywords are extracted from the query to retrieve the documents along with the metadata (as shown in Algorithm 1). Subsequently, data model processing is performed (as described in Algorithm 2). The categories representing web search results are formulated via similarity values. These similarity values are compared with a threshold, i.e., the highest similar document. The Jaccard similarity, i.e., $J(A, B) = A \cap B, A \cup B$, is applied to attain the textual modality $t = Mean(JS) + SD(JS)$ where $t$ is a textual threshold, mean ($JS$) is the mean of Jaccard similarity up to $I$ times and $SD(JS)$ is the standard deviation of Jaccard similarity. The textual similarity involves similarity between document keywords, titles, links, and descriptions, generating a set of $JS$ including all nodes. The reasons for choosing Jaccard similarity specifically in this research are numerous. Firstly, Jaccard similarity is robust to outliers since it does not take the shape of the distributions into account and operates on categorical data, which is important in accurate document hierarchy generation. Secondly, the Jaccard similarity forms the hierarchies based on significant overlapping of the documents with minimal preprocessing requirements (e.g., term frequency/inverse document frequency normalization and vector sparsity issues). This facilitates rapid real-time hierarchy generation. Thirdly, Jaccard similarity works closer in spirit to Boolean search than some other text similarity measures. This is primarily because both Jaccard similarity and Boolean search are based on set operations and binary (yes/no) logic. Hence, considering BU needs, Jaccard similarity was deemed a better choice to form the categorical tree data model based on the Miller cognition rule. The overall flow is outlined in Algorithm 3.

*4.3. Implementation*

A tool-based WSREB mechanism is implemented to allow blind users to explore web search results documents. The tool employed the Django framework and Python libraries via the PyCharm IDE community version 3.7.1, as described in Table 2. The Django web services are utilized for server applications, while the function initiates document and URL mapping. The voice API is exploited for acoustic requests, and Apache PyLucene creates an inverted index of Wikipedia documents.

**5. Evaluations**

We evaluated the WSREB mechanism to analyze the empirical and usability studies. The empirical evaluation assesses the theoretical aspects, whereas the usability evaluation measures the quality and interactivity of the WSREB mechanism-based tool.

*5.1. Empirical Evaluation*

The SERB mechanism aims to facilitate the exploration of the categories linear and non-linearly. This mechanism also provides accessibility in activities of web search results. Precision is utilized to measure the effectiveness of search results. Precision is calculated by the relevant search results divided by the retrieved search results. The efficiency and exploration activities are measured with click-through rates. Hence, the efficiency of the SERB mechanism is measured via precision, and the exploration activities are evaluated through click-through rates. Therefore, in the following section, we discuss the participants, experiment procedure, measures, and experiment results.

5.1.1. Evaluation: Category-based Precision

The empirical evaluation was conducted to measure the efficiency of categories and the exploration activities of the documents. The comparison of the efficiency of the categories is based on the two types, which are voice-based categories and text-based categories. The result is analyzed in a better way. This section depicts the evaluation metric, methodology, and results.

5.1.2. Evaluation Metric

The evaluation metric measures the efficiency of categories through precision and comparison of results based on voice and textual queries. The purpose behind the comparison is that, as the user is blind, the blind user queries through voice, while textual queries through a screen reader require effort. The SERB tool provides a voice query as well as a textual query with interactive categories. The proposed structure for the search results for the blind user provides a better analysis. Measuring query-based precision is a division of relevant documents over the retrieved document search results.

*5.2. Experimental Setup*

The category efficiency measure with the SREB tool is based on retrieved results. The experiments are processed on an Intel(R) Core (TM)i7-4700MQ CPU @ 2.40GHz equipped with 16 GB of RAM and a 64-bit operating system. Textual queries are executed to retrieve the document search results, and data are instantiated in the search results, which are further used in categories and exploration. The formation of the categories is based on the similarity relationship; therefore, the categories show the precision of the relationship as well. The Boolean operators (AND, OR NOT) are also provided for the search. Queries are selected for the comparisons, both voice and textual. The executed queries are based on multiple topics. Table 3 shows the query that is executed. The first keyword depicts the query, and the second keyword depicts the idea or concept of belonging to the first keyword. We conducted multiple experiments, which included five queries to calculate the precision of the textual query-based categories and voice query-based categories. Here, we mentioned that the top 10 (n = 10) results were taken to compute the precision. The *MAPc* was calculated by performing five experiments. Similarly, the Precision *PR*, Average Precision *APv*, and Mean Average Precision (*MAP*) calculated the correctness of the formation of voice-based categories.

**Table 3.** Queries selected for the evaluation.

|  | Operators | Category B |
| --- | --- | --- |
| Animals | AND, OR | Sea animals, land animals, vertebrates, reptiles |
| Corona Virus | AND, OR | Symptoms, cases, countries, vaccine |
| Imran Khan | AND, OR, NOT | Prime minister, cricketer, education |
| Plants | AND, OR | Photosynthesis, sunlight, land |
| Airplanes | AND, OR | Jet fighters, air force |
| Roses | AND, OR | Red, region |
| Sports | AND | Games |
| Wonders of the World | AND | Countries |

### 5.3. Reachability Evaluation

Reachability evaluation is the evaluation of the navigation and exploration activities and the reachability of the results measured via click-through rates (*CTRs*). The computation of the reachability of blind users is based on the queries. The click-through rate serves as a measure of reachability. Hence, reachability is defined as the number of clicks required to search from the source to the destination. The *CTRs* calculate the document search results using voice and textual queries. The path of the *CTRs* is from the source node to the destination node. The formula for the *CTRs* is calculated by Equation (1).

$$RCTR_e \ (|Si{\rightarrow}Di|) = \text{number of clicks} \ (Si{\rightarrow}Di) \tag{1}$$

In this formula, $RCTR_e$ is the reachability via CTRs of the exploration activities, $|Si|$ is represented as the source node, and $|Di|$ is represented as the destination node in the document search results. The average reachability is measured on the set of the query divided by the total number of queries. The formula of the average reachability is measured as:

$$ARCT_{ex} \ (|Si{\rightarrow}Di|) = \sum i =_1{}^n RCT_{ex} \ (|Si{\rightarrow}Di|) \tag{2}$$

Here, the $ARCT_{ex}$ is the average reachability of the exploration activities, $i$ is the ith experiment and $N$ is the total number of the experiments. The $MARCT_{ex}$ is calculated as the number of queries divided by the total number of experiments and calculated by Equation (3) as:

$$MARCT_{ex} \ (|Si{\rightarrow}Di|) = \sum i =_1{}^n RCT_{ex} \ (|Si{\rightarrow}Di|) \tag{3}$$

### 5.4. Usability Evaluation

This questionnaire is utilized to measure the quality of the interface, satisfaction, system usefulness, and information about the system. Nineteen questions are involved to measure the usability, and the scale begins from 1 to 7, which depicts strongly disagreeing to strongly agreeing. The results are shown in Table 4, which summarizes the score of the CUSQ, which covers the interface screen, system information, and terminology, along with the learning and capabilities of the system in a broader range.

**Table 4.** User's CUSQ Evaluation Overall Satisfaction (Overall), System Usefulness (Usefulness), Information Quality (Info. Qua.) and Interface Quality (Inter. Qua.).

| Blind Users | Overall | Usefulness | Info Qua | Inter Qua | Avg | Score |
|:---:|:---:|:---:|:---:|:---:|:---:|:---:|
| BU1 | 5.62 | 5.18 | 5.62 | 5.98 | 5.6 | 0.8 |
| BU2 | 5 | 4.72 | 5 | 5.2 | 4.98 | 0.71 |
| BU3 | 5 | 5.18 | 5.10 | 5.13 | 5.10 | 0.72 |
| BU4 | 5.62 | 5.54 | 6.9 | 7.15 | 6.3 | 0.9 |
| BU5 | 5.37 | 5.18 | 5.1 | 5.8 | 5.3 | 0.75 |
| BU6 | 5.25 | 5.3 | 5.2 | 5.4 | 5.2 | 0.74 |
| BU7 | 4.62 | 4.63 | 5.0 | 5.6 | 4.9 | 0.7 |
| BU8 | 4.37 | 5.1 | 5 | 6 | 5.1 | 0.72 |
| BU9 | 5 | 4.1 | 4.8 | 5.1 | 4.7 | 0.67 |
| BU10 | 3.62 | 3.90 | 4 | 5 | 4.13 | 0.59 |
| BU11 | 4.37 | 4.7 | 5.1 | 5.8 | 4.9 | 0.7 |
| BU12 | 5 | 5 | 5 | 6 | 5.2 | 0.74 |
| BU13 | 5.5 | 4. | 3.89 | 4.5 | 4.6 | 0.65 |
| BU14 | 3.62 | 5.1 | 5.0 | 5.6 | 4.8 | 0.68 |
| BU15 | 5.12 | 5.5 | 5.1 | 5.8 | 5.3 | 0.75 |

**Table 4.** *Cont.*

| Blind Users | Overall | Usefulness | Info Qua | Inter Qua | Avg | Score |
|:---:|:---:|:---:|:---:|:---:|:---:|:---:|
| BU16 | 6 | 5.2 | 5 | 6 | 5.5 | 0.78 |
| BU17 | 5.3 | 5.6 | 5.3 | 5.8 | 5.5 | 0.78 |
| BU18 | 5.08 | 5.3 | 4.8 | 5.1 | 5.0 | 0.71 |
| BU19 | 5.6 | 5.0 | 4.85 | 5.2 | 5.1 | 0.72 |
| BU20 | 5.6 | 5 | 4.9 | 5.6 | 5.2 | 0.74 |
| BU21 | 4.6 | 4.8 | 4.5 | 5 | 4.7 | 0.67 |
| BU22 | 5.5 | 4.8 | 4.5 | 5 | 4 | 0.57 |
| BU23 | 5.2 | 5.6 | 5.1 | 5.8 | 5.4 | 0.77 |
| BU24 | 5.2 | 5.8 | 5.0 | 5.6 | 5.4 | 0.77 |
| BU25 | 6.12 | 5.0 | 4.58 | 5.3 | 5.2 | 0.74 |
| Avg | 5.09 | 5.0 | 4.97 | 5.53 | 5.08 | 0.69 |
| Score | 0.7 | 0.71 | 0.71 | 0.79 | 0.72 | 0.78 |

The score shows the overall usability of the SERB tool is 78%, which is good. The usefulness of the system is 70%, whereas the information quality is 71%. The interface quality is also at 71%. This shows that the overall usability of the SERB tool for blind users is satisfactory.

## 6. Results and Discussions

Figure 3 shows the results presented in the categories based on voice and textual queries. In the first experiment, the voice-based categories were more efficient as compared to text-based categories. In the second experiment, the textual queries showed better results than the voice queries. The third experiment depicted that textual-based categories and voice-based categories show a minimal difference. The fourth experiment shows the results are similar, both with textual-based categories and voice-based categories. The fifth experiment shows that the voice-based categories show more efficient results as compared to the textual-based categories. The calculated results of voice-based categories and text-based categories are presented in Table 5. The conclusion of the five experiments is that the category's $MAPc$ for the textual-based categories is 84 percent, whereas the voice-based categories $MAPc$ is 86%. These results show that blind users can efficiently explore and access the categories with voice. Exploring the categories with voice enables blind users to enhance the search and explore the search results.

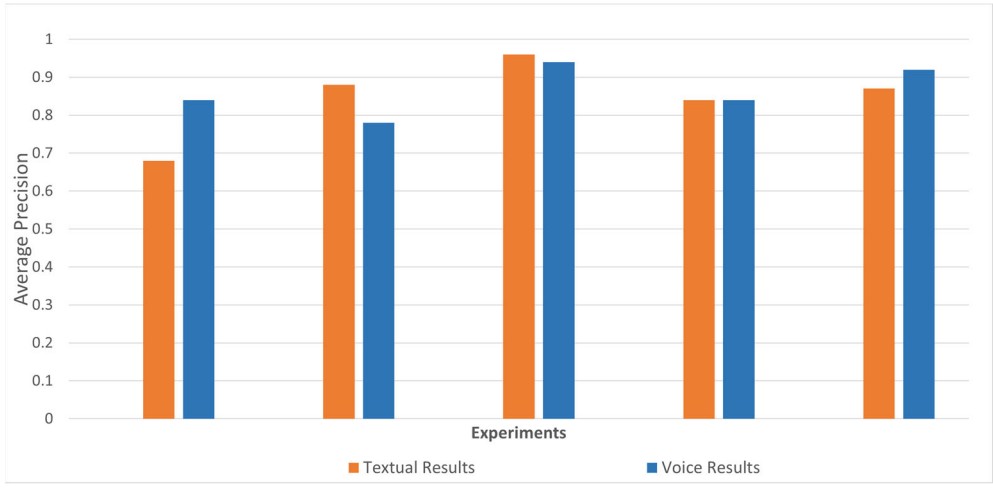

**Figure 3.** Efficiency comparison of categories based on textual and voice search results.

**Table 5.** Precision of categories with textual and voice query results (experiments, Query, Categories' Precision (Pc), Categories' Average Precision (APc), Voice query Precision (Pv), Voice query Average Precision (APv).

| Experiments | Query | $P_c$ | $AP_c$ | $P_v$ | $AP_c$ |
|---|---|---|---|---|---|
| 1 | Qry1 | 1 | | 0.8 | |
| | Qry2 | 1 | | 0.4 | |
| | Qry3 | 1 | 0.68 | 1 | 0.84 |
| | Qry4 | 0 | | 1 | |
| | Qry5 | 0.4 | | 1 | |
| 2 | Qry1 | 1 | | 1 | |
| | Qry2 | 1 | | 1 | |
| | Qry3 | 0.8 | 0.88 | 0.6 | 0.78 |
| | Qry4 | 0.6 | | 1 | |
| | Qry5 | 1 | | 0.3 | |
| 3 | Qry1 | 1 | | 1 | |
| | Qry2 | 1 | | 1 | |
| | Qry3 | 1 | 0.96 | 1 | 0.94 |
| | Qry4 | 0.84 | | 0.90 | |
| | Qry5 | 1 | | 0.82 | |
| 4 | Qry1 | 1 | | 1 | |
| | Qry2 | 0.4 | | 0.43 | |
| | Qry3 | 0.8 | 0.84 | 1 | 0.846 |
| | Qry4 | 1 | | 1 | |
| | Qry5 | 1 | | 0.8 | |
| 5 | Qry1 | 1 | | 0.90 | |
| | Qry2 | 1 | | 1 | |
| | Qry3 | 0.75 | 0.876 | 1 | 0.928 |
| | Qry4 | 0.63 | | 0.74 | |
| | Qry5 | 1 | | 1 | |
| MAP | | | 0.84 | | 0.865 |

The reachability is built on the *CTR* results taken from the source node to the destination node against the query. The raking list is considered from the source node to the destination node of the document search results. Figure 4 shows the average results of the CTR for each experiment that is conducted. The *ARCTR* of the experiments is 6.5, 7, 6.9, 6, and 7.9 to reach the results of the documents at the position that is defined by the exploration mechanism of the SERB tool. The results of the *MARCTRe* of all experiments in numbers are 6.86. It depicts that approximately seven clicks are used to reach the destination results at a certain ranking position. The *ARCTR* of the experiments is 5, 6, 5.9, 6.5, and 6 to reach a certain position of the documents defined by the mechanism of the SERB tool. The *MARCTR* of the experiment is 5.88, which means five clicks are required to reach the destination results of the documents at a certain ranking position. The results depict that the voice query is efficient and reduces the number of clicks that are feasible for reaching the destination. Hence, the proposed approach presents various implications, as provided in Table 6 for BUs. These applications demonstrate the versatility and potential impact of AI-driven content summarization in enhancing BUs' access to information across various domains. By automating the process of distilling essential information, this technology can empower blind BUs to navigate, understand, and engage with online content more effectively.

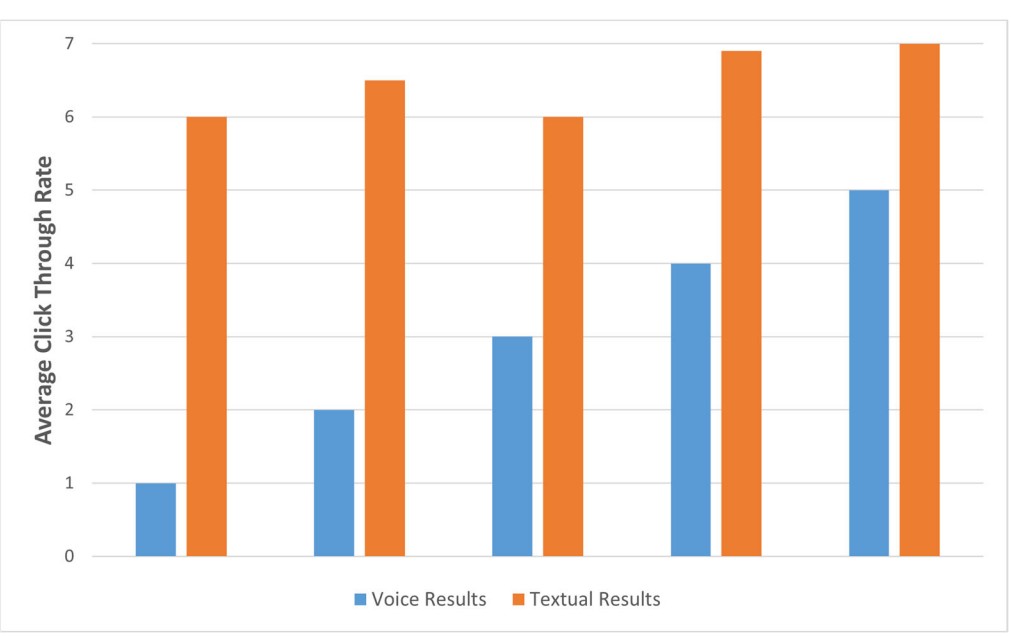

**Figure 4.** Query-based reachability results.

**Table 6.** Potential applications of AI-driven content summarization for BUs.

| Application | Description |
| --- | --- |
| Web Content Summarization | Automatically generate concise summaries of web articles, blog posts, and news articles, aiding BU efficient content consumption. |
| Academic Material Summarization | Summarize lengthy academic papers, research articles, and textbooks, enabling BUs to grasp key concepts and findings without reading every detail. |
| Document Summarization | Generate summaries for various document types, including contracts, legal documents, and manuals, making complex information more accessible to BUs. |
| Email and Communication | Summarize lengthy emails, communication threads, and documents shared electronically, allowing BUs to manage correspondence more effectively. |
| News and Updates Aggregation | Automatically extract essential details from multiple news sources, presenting BUs with concise and timely updates on current events and topics. |
| Educational Content Summarization | Summarize educational videos, lectures, and online courses, helping BUs grasp the main concepts and lessons without needing to watch or read the entire content. |
| Navigational Assistance | Summarize navigational instructions, maps, and directions, providing BUs with concise guidance for travel and navigation. |
| Technical Documentation | Condense technical manuals, user guides, and documentation for software and hardware, aiding BUs in understanding and troubleshooting complex systems. |
| Social Media and Posts | Summarize lengthy social media posts, threads, and comments, enabling BUs to engage with online discussions more efficiently. |
| Personalized News Feeds | Provide BUs with tailored news summaries based on their interests and preferences, helping them stay informed without being overwhelmed by content. |

*6.1. Computational Analysis*

The WSREB mechanism involves several components, including querying, document retrieval, grouping, hierarchical and categorical data modeling, and interface interactions. Each of these components may have its own computational complexity and calculated as follows:

- Querying and document retrieval might have a complexity of $O(n)$ where n is the number of documents retrieved.
- Grouping and statistical methods could contribute additional complexity.

- ○ Hierarchical and categorical data modeling might involve tree traversals, which could be O(log n) for balanced trees, or even O(n) in the worst case if not properly optimized.
- ○ The interface interactions may have constant time complexity (O(1)). Considering the interplay of these components, the overall complexity of the WSREB mechanism could be quite complex and not easily reducible to a single Big O notation.

*6.2. Theoretical and Practical Implications of the Framework*

Traditionally, the existing information systems presented the search results as ranked snippets that compel BUs to sift through information sequentially, resulting in time-consuming searches and potential disorientation. The struggle intensifies due to the deficiency of navigational aids designed with BUs in mind. To overcome this challenge, we proposed a theoretical framework that categorically organized the search results space as hierarchies. To determine the effectiveness of the proposed framework, we conducted a thorough empirical and usability evaluation yielding satisfactory results. The proposed approach provided promising results in allowing BUs to effectively browse the relevant information and with each step get closer to the required information logarithmically. The implications of the proposed approach are diverse, especially when augmented with Artificial Intelligence (AI).

Potential applications of AI-driven content are summarized for BUs in Table 6. Specifically, instead of integrating the search results, the proposed framework can incorporate AI-based summarized documents. This especially can aid academic users in summarizing and organizing the academic literature. While the traditional information system can be challenging for BUs to decipher using screen readers, potentially leaving them with an incomplete understanding of the webpage's content, the proposed framework allows BUs to formulate their search journey and navigate via voice commands.

Assistive technologies, such as screen readers and voice assistants, are available to aid BUs in navigating the web. However, not all websites are compatible with these tools, and their limitations may hinder a seamless experience. The cumulative cognitive load of listening to synthesized speech, processing information, and navigating can lead to mental fatigue for BUs. In response to these challenges, we devised our approach conforming to the cognitive needs of the BUs. Ultimately, the goal is to create a digital environment that accommodates the diverse needs of individuals, ensuring equal access to information and participation for all, regardless of visual ability.

## 7. Conclusions

In conclusion, the present era mainly relies on web search as the primary means of accessing information. Within this context, blind users (BUs) constitute a significant portion of web users. Despite recent technological advancements in web search mechanisms, BUs continue to face difficulties in accessing online services, leading to approximately 53% success of the search sessions. These challenges can be attributed to inherent shortcomings in information organization mechanisms on the web. Notably, BUs interact with information linearly, having to sequentially determine the relevance of content, resulting in cognitive strain and time consumption, ultimately leading to information disorientation. Therefore, in this research, we investigated a non-linear information exploration mechanism for BUs. The categorical data model interacts with the tree data model to generate hierarchies based on the similarity threshold. We leveraged the multimodal (textual and voice) interaction of the BUs for searching on the web using the Wikipedia dataset. The efficacy of the proposed mechanism was evaluated from empirical and usability perspectives. The empirical evaluation showed 84% and 86.5% for the categorical precision and voice query precision, respectively. The behavioral analysis showed the accessibility of the search results within five clicks on average. Table 7 describes the potential future directions of the proposed system.

**Table 7.** Potential future directions for AI-driven content summarization.

| Future Work | Description |
|---|---|
| Enhanced Abstractive Summarization | Develop more advanced abstractive summarization techniques that can generate summaries with higher coherence and readability. Incorporate contextual understanding and style mimicry. |
| Customizable Summary Length | Allow users, including blind users, to specify the desired length of the summary based on their preferences and reading capabilities. |
| Multilingual Summarization | Extend AI-driven summarization to support multiple languages, enabling blind users to access content in their preferred language. |
| Domain-Specific Summaries | Create specialized summarization models for various domains (e.g., scientific literature, news, legal documents) to cater to diverse informational needs. |
| Adaptive Summarization | Develop algorithms that adapt summarization based on user feedback, continually improving the quality of generated summaries for blind users. |
| Real-time Summarization | Implement summarization techniques that can generate summaries in real-time, enabling immediate access to key information as blind users navigate the web. |
| Integration with Assistive Tools | Integrate AI-generated summaries seamlessly with screen readers and other assistive technologies commonly used by blind users. |
| Cross-Modal Summarization | Explore generating summaries not only in text but also in alternative formats, such as audio summaries, to accommodate varying accessibility preferences. |
| Evaluation with Blind Users | Conduct thorough user studies and evaluations involving blind users to assess the effectiveness, usability, and impact of AI-driven content summarization. |
| Privacy-Aware Summarization | Develop techniques that generate accurate summaries while respecting user privacy, ensuring sensitive content is not exposed in the summary. |
| Hybrid Approaches | Combine extractive and abstractive summarization methods to leverage the strengths of both approaches for improved accuracy and readability. |

The usability evaluation covering the interface screen, system information, and terminology, along with the learning and capabilities of the system in a broader range, showed an overall 72.5% usability score. While this research focused on architectural aspects of BU web exploration, in the future, we are interested in investigating the effectiveness of the proposed approach in various instantiation tools such as deep learning and in a standalone service environment that can be embedded in voice-activated assistive technologies. Furthermore, we are also interested in performing a comparative analysis of the existing BU information exploration assistive tools for a detailed investigation of BU information exploration behavior.

**Author Contributions:** Conceptualization, S.N., U.R., M.S., A.R.K., Q.A. and Y.D.; Data curation, S.N., U.R., M.S. and Y.D.; Formal analysis, U.R., M.S., A.R.K. and Q.A.; Funding acquisition, U.R., Q.A. and Y.D.; Investigation, S.N., M.S. and Y.D.; Methodology, S.N., U.R., M.S., A.R.K., Q.A. and Y.D.; Project administration, Q.A.; Resources, M.S., Q.A. and Y.D.; Software, S.N. and Y.D.; Supervision, U.R. and A.R.K.; Validation, U.R., M.S. and A.R.K.; Visualization, U.R., A.R.K. and Q.A.; Writing—original draft, S.N., U.R., M.S., A.R.K., Q.A. and Y.D.; Writing—review and editing, U.R., A.R.K., Q.A. and Y.D.. All authors have read and agreed to the published version of the manuscript.

**Funding:** This work was supported and funded by the Deanship of Scientific Research at Imam Mohammad Ibn Saud Islamic University (IMSIU) (grant number IMSIU-RP23122).

**Institutional Review Board Statement:** Not Applicable.

**Informed Consent Statement:** Not Applicable.

**Data Availability Statement:** The datasets generated and/or analyzed during the current study are available from the corresponding author upon reasonable request.

**Acknowledgments:** This work was supported and funded by the Scientific Research at Imam Mohammad Ibn Saud Islamic University (IMSIU) (grant number IMSIU-RP23122).

**Conflicts of Interest:** The authors declare no conflict of interest.

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
