# Peer review of "WSREB Mechanism: Web Search Results Exploration Mechanism for Blind Users"

_applsci, doi:10.3390/app131911007_

Round 1

Reviewer 1 Report

The novelty of the proposed paper is that it proposes a new Web Search Results Exploration Mechanism (WSRM) that is designed specifically for users with visual impairments (BUs). The proposed WSRM addresses the shortcomings of existing WSRMs by providing a more interactive experience for BUs.

The research paper is well-organized and follows the template of the Applied Science Journal. The introduction section provides an overview regarding Blind Users web search queries. The proposed research aims are presented within this section. The article makes use of an information exploration mechanism for BUs that considers the BUs' cognitive factors and categorically organizes the search result space for enhanced access.

The literature review section is divided into four well defined subsections. Each subsection integrates a good amount of recently published research papers and associated references.

The following section presents the WSREB-Mechanism which elaborates formal representations commencing tree-based data models and categorical data models. It also uses component-based architecture and Simple and Usable Interface (SUI) design. This mechanism emphasizes accessibility and navigation of web documents while reducing the cognition load in a nonlinear and integrated way. The proposed mechanism approach, algorithms, definitions, and component-based architectural representations are detailed within this section.

The proposed research makes use of the popular Wikipedia datasets, which I consider to be an excellent dataset to be benchmarked as this is a large and comprehensive dataset that contains a bulk of web documents covering a diverse range of domains and topics. This makes it a valuable resource for the proposed exploration mechanism.

The evaluation section of the paper integrates both empirical and usability studies, which are important to validate the proposed exploration mechanism. The empirical studies assess the effectiveness of the mechanism in helping users to explore web search results, while the usability studies measure the ease of use and user satisfaction with the mechanism.

The conclusions of the paper are based on the research findings. These were obtained from the empirical and usability studies that were conducted to evaluate the proposed mechanism. The empirical studies showed that the mechanism is effective in helping BUs to explore web search results. The usability studies showed that the mechanism is user-friendly and that users are satisfied with the results.

The future works proposed by the authors are aimed to further improve the existing WSRMs aimed to enhance BUs experience.

Author Response

Original Article Title: WSREB Mechanism: Web Search Results Exploration Mechanism for Blind Users

To: Editor in Chief,

MDPI, Diagnostics

Re: Response to reviewers

Dear Editor,

Many thanks for insightful comments and suggestions of the referees. Thank you for allowing a resubmission of our manuscript, with an opportunity to address the reviewers’ comments.

We are uploading (a) our point-by-point response to the comments (below) (response to reviewers), (b) an updated manuscript with green, blue, and orange highlighting indicating changes, and (c) a clean updated manuscript without highlights (PDF main document).

By following reviewers’ comments, we made substantial modifications in our paper to improve its clarity, English and readability. In our revised paper, we represent the improved manuscript such as:

(1) Revised Abstract, (2) Revised Introduction, (3) Results section, (4) Discussions and Conclusion sections.

We have made the following modifications as desired by the reviewers:

Best regards,

Corresponding Author,

Dr. Qaisar Abbas (On behalf of authors),

Professor.

Reviewer 2 Report

The paper presents a web search mechanism designed to facilitate blind users in information retrieval. The authors applied a multimodal interaction interface to ease the information narrowing procedure. The research topic is interesting, which may lead to practical applications. However, the algorithm design lacks novelty. More sophisticated methods should be considered to improve the work and the performance of the system. For example, while calculating the matching score between the query and the documents, the authors used Jaccard similarity which is way too simple. For information retrieval and text categorization, they should at least consider the classical vector space model and use tf-idf for computation. Also, while constructing the hierarchical tree of the documents, they partitioned the documents disjointly according to different sub-categories. In the real world, however, a document can belong to multiple categories. According to their design, if a blind user searches through a wrong branch, he can easily miss the correct document(s) even though the document(s) have been retrieved. Please study "topic models" in text mining and consider improving the design.

In the paper, the authors focused on presenting the evaluation of their own method. They should also consider comparing the performance of their design with the existing ones in assisting blind users, and then analyze the pros and cons, which will make the research findings more convincing and valuable.

Writing issues exist throughout the paper. Below are some examples from the first eight pages:

(1) line 36, "manne [5]r" should be "manner [5]"
(2) lines 36-37, "However, despite Blind Users (BUs) comprising 75 million of the population, face significant challenges, hindering their ability to access and comprehend web content [7]", awkward wording !
(3) line 56, "discusses the state-of-the-art", missing a noun at the end !
(4) line 168, "Although, the tool facilitates all visually impaired users", incorrect use of "although", either combine the sentence with the preceding one or change it to "The tool facilitates all visually impaired users though". The second way is simpler but kind of informal.
(5) lines 180-182, "While users with ambiguous search goals conduct an exploratory search to explore and navigate through
the retrieved results, as well as refine their query as needed." incomplete sentence !
(6) line 219, "query q retrieves a document d which contains relevant textual information against q", why "against" q? It should be related to q, right?  Please consider deleting "against q" to fix the meaning.
...

Proofreading is required if the authors plan to re-submit their paper.

Author Response

(The authors gave the same response as above.)

Reviewer 3 Report

Good research work. Only minor format changes, such as avoiding white lines on page 14 or improving an image resolution.

English seems of a good quality for non-native English readers

Author Response

(The authors gave the same response as above.)

Reviewer 4 Report

The author's interest in his work is to present a solution for web search and information retrieval from the internet for blind users. The authors proposed a framework based on the Wikipedia dataset and categorized the search results. The proposed framework from empirical and usability perspectives evaluated on legal Bus.

The paper well-constructed.

The authors presented a good overview of the state of the art.

The paper in its current state can be considered for publication, yet some changes need to be carried out for the final version.

·         In the author names section, the author should revise the ORCID appearances, according to the journal template.

·         I detected a typo in line 36 between word and reference (manne [5]r).

·         The definition presented on page 8 should be presented as defining the algorithm …

·         In subsection 5.1 the authors give the current challenges of blind users which I do not understand the purpose of presenting at the end of the paper! Should it be presented in the previous sections (introduction for example)?

·         The authors should enhanced the results in Figures 3 and 4 quality they look blurred.

·         In Figure 4 definition missing what blue and orange refer to.

·         I recommend the authors to remove spaces in reference pages 18 and 19.

·     The experiment section should be more detailed, about why the authors chose this solution compared with deep and transfer learning.

Author Response

Original Article Title: WSREB Mechanism: Web Search Results Exploration Mechanism for Blind Users

To: Editor in Chief,

MDPI, Applied Sciences

Re: Response to reviewers

Dear Editor,

Many thanks for insightful comments and suggestions of the referees. Thank you for allowing a resubmission of our manuscript, with an opportunity to address the reviewers’ comments.

We are uploading (a) our point-by-point response to the comments (below) (response to reviewers), (b) an updated manuscript with green, blue, and orange highlighting indicating changes, and (c) a clean updated manuscript without highlights (PDF main document).

By following reviewers’ comments, we made substantial modifications in our paper to improve its clarity, English and readability. In our revised paper, we represent the improved manuscript such as:

(1) Revised Abstract, (2) Revised Introduction, (3) Results section, (4) Discussions and Conclusion sections.

We have made the following modifications as desired by the reviewers:

Best regards,

Corresponding Author,

Dr. Qaisar Abbas (On behalf of authors),

Professor.

Reviewer 5 Report

This study addresses the challenges blind users (BUs) face when using web search services. BUs often find accessing online information difficult due to the linear nature of information interaction and the inadequacy of standard web service evaluation metrics. The study introduces a new framework, tested on a group of 25 legally blind individuals, which categorizes search results based on cognitive principles guiding BUs. Results show an 84% accuracy in categorization and a usability score of 72.5%, promising improved web content accessibility and user experience for BUs.

The paper is intriguing, and its impact on the field could be substantial. I understand that the paper has already undergone at least one round of revision. Nonetheless, the current state shows some aspects that should be addressed to improve the paper's quality. These aspects, along with some comments, are provided below:

  • Major contributions in the Introduction should be expanded; I suggest the authors to expand each point of the list.
  • The authors made a solid effort in highlighting the limitations of the related literature. In light of this, the limitations of the proposed study should be provided also.
  • A graphical depiction of the workflow of the suggested method could be provided, along with a brief explanation of it.
  • I appreciated the Results and Discussions. The authors could improve this section by discussing theoretical and practical implications of this study in a more comprehensive manner.
  • Figures are coarse grained. I suggest to provide them in a higher-quality format.
  • There are several presentation errors. For example, Table 4 overlaps with line numbers. Please fix those.

Author Response

(The authors gave the same response as above.)

Round 2

Reviewer 2 Report

The re-submission didn't show much improvement (in fact, more format problems occurred). There are still technical issues:

As for the concern "while constructing the hierarchical tree of the documents, the authors partitioned the documents disjointly according to different sub-categories. In the real world, however, a document can belong to multiple categories. According to their design, if a blind user searches through a wrong branch, he can easily miss the correct document even though it has been retrieved.", the authors responded with "To overcome the overlapping issue, we used k-array data structure. Although this is rare to happen, if two documents overlap, they are created as a third child of the parent in the hierarchy." The answer shows that the authors didn't even understand the question which is not related to "two or more similar documents". The concern is that they should not partition the documents disjointly since a document can belong to multiple categories and therefore should appear in multiple branches. This is very common in practice, not rare. For example, an article talking about football players can belong to both "sports" and "business". Again, the authors are recommended to study the "topic models" in text mining and then improve their design.

The authors said that they used Jaccard similarity because of the data sparsity. However, Jaccard similarity is just one method for measuring the closeness between the data. It doesn't solve the sparsity problem itself. The concern is actually about the accuracy of using Jaccard similarity since it is binary and does not consider the importance of different terms (e.g., tf-idf is way better regarding this, and cosine similarity is more frequently used in measuring the similarity of text data). For sparsity issue, the authors should consider other approaches to overcome the problem (e.g., PCA)

Section 3.2 has a logical problem in the definition. In line 288, ? = {?1, d2. . . , d?}, and in line 295, it says ? ? {??, ??, ??}. How can D be a set of documents and later on belongs to the individual documents? While doing research, please take it seriously. All the definitions should be re-examined to ensure that the concepts are clear and accurate.

lines 66 to 81 present a summary of the contribution and the paper structure. It is unnecessary to do them in subsections. The first version is better and what people do typically at the end of introduction.

The resolution of Figures 1 and 2 is very low. In the first version, the problem was not there. I don't know why the authors made the change and presented them in low quality. Figures 3 and 4 also need improvement in resolution. All four figures are unnecessarily wide. Their sizes in the first version are better.

Tables 4 and 6 are way too wide and overlap with the line numbers. This problem was not there in the first version. The other tables except 5 are also wide. Actually, all the Tables were better presented in the first version. The authors should recover the original presentation style.

All hyperlinks connecting the citations to the references were removed. If this is not required by the journal, the authors should add them back, which does not hurt the presentation and is more convenient for the readers to track the references. Overall, the first version's format is much better than the second regarding figures, tables, and citations.

There are still many writing issues. For example:

(1) line 12, "web search functions as a pivotal conduit for information dissemination". Grammar error (missing a verb), not a complete sentence
(2) line 21, "Addressing this, our study introduces ...", it is better to change it to "To address the problem, we introduce"
(3) lines 25 to 26, remove the commas before and after "conducted on a cohort of legally blind individuals (N = 25)". It is better not to break the sentence too many times for smooth reading.
(4) line 45, "which, while gauging relevance, fail to encapsulate subjective user satisfaction, particularly ...", should be "which fail to encapsulate subjective user satisfaction while gauging relevance, particularly ...", again, do not break the sentences too many times in formal writing.
(5) line 287, "A set of multimodal queries Q containing multiple queries ? = {?1, q2. . . , q?} generates multiple documents D formulate a set of documents therefore ? = {?1, d2. . . , d?}." awkward and unclear writing. break it into two or three short sentences if needed.
(6) line 289, delete "further"
(7) line 291, "... forms a set, therefore, ? = ?1, k2. . . , k?", "forms" should be "form", the word "therefore" is unnecessary, and it is better to use a set symbol, e.g., "... form a set ? = {?1, k2. . . , k?}".
(8) line 292, "Moreover, formation of ? = ..." simplify to "And ? = ..."
(9) line 301, "A document tree DT is an also tree", awkward wording
(10) line 209, "Now again γ is applied on hns to calculate the similarity between Q and the parent node making it â„Ž???′ then arranging it in descending order", delete "Now again" and change "ns" of "hns" to subscript. Also, what do you mean "making it â„Ž???′ then arranging it in descending order". The sentence is very awkward.

Please note that these examples are only selected ones. It is not the reviewer's job to help the authors polish their writing. Please proofread the paper seriously and thoroughly.  

Author Response

Dear Reviewer,

We have improved the paper as suggested by you. The attached response letter is for further updation as requested by you.

Thank you very much.

Reviewer 4 Report

The author's interest in his work is to present a solution for web search and information retrieval from the internet for blind users. The authors proposed a framework based on the Wikipedia dataset and categorized the search results. The proposed framework from empirical and usability perspectives evaluated on legal Bus.

The paper well-constructed.

The authors presented a good overview of the state of the art.

After modification made by the authors, the paper in its current state can be considered for publication.

Author Response

Reviewer#5

The authors successfully addressed all my concerns.

Author response: Thank you for appreciating our hard work.

Reviewer 5 Report

The authors successfully addressed all my concerns.

Author Response

(The authors gave the same response as above.)
